# The Functionalization of PES/SAPO-34 Mixed Matrix Membrane with [emim][Tf$_2$N] Ionic Liquid to Improve CO$_2$/N$_2$ Separation Properties

Jonathan S. Cardoso [1,2,3,4,*] ID, Zhi Lin [2] ID, Paulo Brito [3,4] ID and Licínio M. Gando-Ferreira [1,*] ID

[1] CIEPQPF, Department of Chemical Engineering, Faculty of Sciences and Technology, University of Coimbra, Pólo II, Rua Sílvio Lima, 3030-790 Coimbra, Portugal

[2] CICECO, Department of Chemistry, University of Aveiro, 3810-193 Aveiro, Portugal; zlin@ua.pt

[3] Mountain Research Centre (CIMO), Polytechnic Institute of Bragança, Campus de Santa Apolónia, 5300-253 Bragança, Portugal; paulo@ipb.pt

[4] Associate Laboratory for Sustainability and Technology in Mountains Regions (SusTEC), Polytechnic Institute of Bragança, Campus de Santa Apolónia, 5300-253 Bragança, Portugal

[*] Correspondence: jonathancardoso@ipb.pt (J.S.C.); lferreira@eq.uc.pt (L.M.G.-F.)

**Abstract:** The use of ionic liquid [emim][Tf$_2$N] as an additive in polyethersulphone (PES) and nano-sized silico-aluminophosphate-34 (SAPO-34) mixed matrix membrane was studied through the incorporation of different amounts of [emim][Tf$_2$N] in the membrane composition, as presented in this work, varying from 10 to 40 wt%. Through gas permeation tests using CO$_2$ and N$_2$, the membrane composition containing 20 wt% [emim][Tf$_2$N] led to the highest increase in CO$_2$ permeability and CO$_2$/N$_2$ selectivity. The use of low concentrations of additive (10–20 wt%) promoted a state called antiplasticization; in this state, the permeability was even more regulated by the kinetic diameter of the species which, in this work, permitted achieving a higher CO$_2$/N$_2$ selectivity while increasing the CO$_2$ permeability until an optimal condition. [emim][Tf$_2$N] also promoted a better dispersion of SAPO-34 particles and an increase in the flexibility of the polymeric matrix when compared to a film with the same composition without [emim][Tf$_2$N]. Moreover, the characterizations corroborated that the inclusion of [emim][Tf$_2$N] increased the zeolite dispersion and improved the polymer/zeolite compatibility and membrane flexibility, characterized by a decrease in glass transition temperature, which helped in the fabrication process while presenting a similar thermal resistance and hydrophilicity as neat PES membrane, without affecting the membrane structure, as indicated by FTIR and a contact angle analysis.

**Keywords:** gas separation; mixed matrix membranes; ionic liquid; polyethersulfone; SAPO-34

## 1. Introduction

Pressure Swing Adsorption (PSA), Temperature Swing Adsorption (TSA), and cryogenic distillation are industrially used in CO$_2$ separation. However, those processes require a high energy demand and construction area, solvent post-treatment, and complex design; with this, the use of membranes in gas separation continues to be an attractive process, since it provides a lower energy consumption, is modulable, requires smaller processing area, does not require the regeneration of solvents, and can be connected to traditional processes. Mixed matrix membranes have a wide combination of polymers and fillers, showing a permeability and selectivity above the Robeson upper limit. Polyethersulfone (PES), polyimide (PI), polysulfone (PSf), and Matrimid 5218 are the common polymer matrixes, while zeolite 4A, carbon molecular sieves, and SAPO-34 are the most common fillers. Silicoaluminophosphate-34 (SAPO-34) is used for gas separation because of its pore size (0.38 nm), which is near the kinetic diameter of gases like H$_2$ (0.29 nm), CO$_2$ (0.33 nm), N$_2$ (0.36 nm), CO (0.37 nm), and CH$_4$ (0.38 nm). The incorporation of an inorganic filler in

a polymeric matrix is intended to increase thermal stability and selectivity when compared to neat polymeric membranes, while keeping the characteristics of organic membranes in terms of high permeability and versatility [1–8].

However, the use of fillers in mixed matrix membranes presents a drawback. The compatibility between the polymeric matrix and the filler material is crucial. Some fillers may chemically interact with the polymer, causing degradation or undesired chemical reactions. Moreover, achieving a good dispersion and adhesion of the filler within the polymer matrix is essential. Poor adhesion can result in the formation of voids or weak interfaces, reducing the effectiveness of the filler in enhancing the membrane properties. The compatibility between the inorganic dispersed material and the organic dispersive media can lead to interfacial voids between the polymer and zeolite, creating non-selective defects that lower the separation performance of the membrane [9–14]. The use of different methods to improve the polymer/filler interaction are reported in the literature, with surface modification being the most common. It is intended to modify the filler surface to enhance compatibility with the polymer by adding functional groups or coatings to minimize chemical reactions; as presented in Table 1, several studies intend to use different surface modifiers to improve the separation performance in $CO_2/CH_4$ and $CO_2/N_2$. The use of additives in the membrane composition should improve the zeolite–polymer matrix interaction, increase dispersion, and improve separation performance [15–17].

**Table 1.** Comparison of $CO_2$ permeability or permeance, $CO_2/CH_4$ and $CO_2/N_2$ selectivities in the literature.

| Filler | %Filler | Polymer | $P_{CO2}$ | $\alpha_{CO2/CH4}$ | $\alpha_{CO2/N2}$ | Reference |
|---|---|---|---|---|---|---|
| Neat | - | PES | 2.6 barrer | - | 18.57 | [18] |
| Neat SAPO-34 (stainless steel) | - | - | 328 GPU | 159 | 29 | [19] |
| Neat SAPO-34 (porous alumina) | - | - | 892 GPU | - | 7.09 | [20] |
| Neat SAPO-34 (silica) | - | - | 6000 GPU | - | 53 | [21] |
| Neat | - | PES | 6.7 barrer | 37.8 | 34.04 | |
| SAPO-34 | 20 wt% | PES | 8.25 barrer | 42.6 | 34.51 | [6] |
| SAPO-34 | 30 wt% | PES | 8.9 barrer | 48.3 | 7.08 | |
| Neat | - | PES | 4.45 barrer | 33.2 | - | |
| HMA 10% | - | PES | 0.8 barrer | 32.3 | - | [22] |
| SAPO-34 | 20 wt% | PES | 5.7 barrer | 37 | - | |
| SAPO-34 + HMA 10% | 20 wt% | PES | 1.3 barrer | 44.7 | - | |
| HMA 4% | - | PES | 5.1 barrer | 39.3 | - | |
| SAPO-34 | 20 wt% | PES | 13.8 barrer | 32.7 | - | [15] |
| SAPO-34 + HMA 4% | 20 wt% | PES | 7.8 barrer | 41.6 | - | |
| SAPO-34 | 20 wt% | PES | 85.69 GPU | 20.67 | - | |
| SAPO-34 + IL 5% | 20 wt% | PES | 230.81 GPU | 46.20 | - | |
| SAPO-34 + IL 10% | 20 wt% | PES | 255.69 GPU | 58.83 | - | [23] |
| SAPO-34 + IL 15% | 20 wt% | PES | 279.26 GPU | 60.62 | - | |
| SAPO-34 + IL 20% | 20 wt% | PES | 300 GPU | 62.58 | - | |
| SAPO-34 | 20 wt% | PES | 30 GPU | 1.3 | - | |
| SAPO-34 + m-EDA | 20 wt% | PES | 10 GPU | 12.14 | - | [12] |
| SAPO-34 | 20 wt% | PES | 50 GPU | 2.5 | - | |
| SAPO-34 + IL | 20 wt% | PES | 0.03 GPU | 4.9 | - | |
| SAPO-34 + m-EDA + IL | 20 wt% | PES | 0.09 GPU | 26.5 | - | [24] |
| SAPO-34 + m-HA + IL | 20 wt% | PES | 0.045 GPU | 37.2 | - | |

Due to its dual nature, ionic liquids can help to enhance the interaction in a mixed matrix membrane. [emim][Tf$_2$N] is intended as one of ionic liquid additives to membranes since it presents high $CO_2$ adsorption, is chemically stable, presents a low melting point,

and a cation and anion composed of organic structures, allowing it easy interaction with organic and inorganic compounds [14,23–28].

This work aims to study the influence of different concentrations of [emim][Tf$_2$N] varying between 10 and 40 wt%, in polymer mass, in a fixed composition of PES/SAPO-34 mixed matrix membrane. Ideal $CO_2$ and $N_2$ permeabilities and $CO_2/N_2$ selectivities are obtained and compared to reach an optimal condition. The neat PES, mixed matrix PES/SAPO-34 and modified mixed matrix PES/SAPO-34/[emim][Tf$_2$N] were characterized by assets of its structure, thermal properties, and hydrophilicity, which are vital to explain how the use of [emim][Tf$_2$N] can interfere with membrane characteristics and performance.

The addition of different amounts of [emim][Tf$_2$N] led to an optimal condition, 20 wt% of [emim][Tf$_2$N], without compromising the MMM general structure and characteristics. The inclusion of a plasticizer additive in MMM preparation promoted an antiplasticization phase in low additive concentrations, between 10 and 20 wt%, resulting in a higher $CO_2/N_2$ selectivity when adding 20 wt% of [emim][Tf$_2$N] than other modified PES/SAPO-34 membranes. However, the permeability of all species is lower than the ones in the literature for $CO_2/N_2$ separation. This fact may be derived from the increase in filler/polymer compatibility since the reduction in interfacial voids leads to lower permeabilities while increasing selectivity.

## 2. Results and Discussion

### 2.1. SAPO-34 Synthesis and Characterization

Figure 1 presents the XRD of the synthesized SAPO-34 sample in a 24 h reaction time and compared to a Chabazite (CHA) standard to verify the purity of the sample obtained. CHA is considered to be the standard for SAPO-34 XRD characterization by the International Zeolite Association (IZA). Both CHA and SAPO-34 present the same tetrahedral structure coordinated with four oxygen, each of which forms a bridge between neighbouring tetrahedral atoms, the combination of which results in a zeolite structure with the same planes of dispersion.

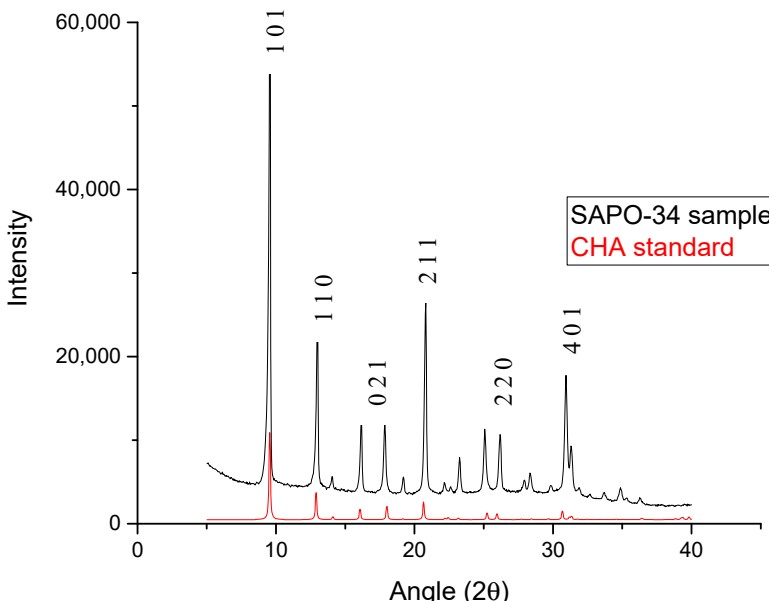

**Figure 1.** XRD for SAPO-34 sample and CHA standard.

SAPO-34 was synthetized using the methodology described in Section 2.2. The resulting SAPO-34 sample presented a high purity in a 24 h reaction time according to the CHA standard diffractogram used as comparison criteria; this methodology follows the proceedings of the International Zeolite Association (IZA)

A low Si/Al molar ratio ($\leq 0.15$) leads to the formation of other structures (SAPO-5 or AlPO-34) due to Si content not being enough to substitute the Al and P atoms in the

tetrahydric structure. However, a high Si/Al content leads to a low $CO_2$ adsorption [21,29]. Table 2 presents that the XRF analysis revealed a Si/Al ratio of 0.29 for the post-synthesized SAPO-34 sample.

**Table 2.** Si/Al molar ratio in original gel and crystal samples.

| Powder | Si/Al Molar Ratio | |
|---|---|---|
| | **Gel** | **Crystal** |
| SAPO-34 sample | 0.30 | 0.29 |

The adsorption–desorption nitrogen isotherm is presented in Figure 2 and the textural properties of the zeolite in Table 3 that allow for a better understanding of the zeolite characteristics according to the porous size and total area.

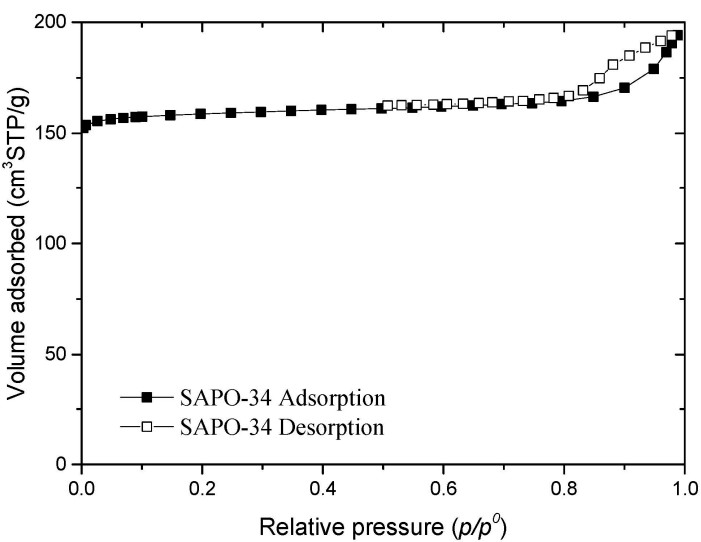

**Figure 2.** Adsorption–desorption nitrogen isotherms for SAPO-34 sample.

**Table 3.** Textural properties of the materials.

| Sample | $S_{BET}$ (m² g⁻¹) | $S_{Langmuir}$ (m² g⁻¹) | $S_{ext}$ (m² g⁻¹) | $S_{mic}$ (m² g⁻¹) | $V_{mic}$ (mm³ g⁻¹) |
|---|---|---|---|---|---|
| SAPO-34 | 464 | 707 | 9 | 455 | 0.249 |

The adsorption–desorption nitrogen isotherms presented in Figure 3 show a type I isotherm which is characteristic for microporous materials such as SAPO-34. The extent of adsorption increases with pressure until saturation is reached, at which point, no further adsorption occurs. The presence of a small hysteresis at a high relative pressure can also be noted, indicating a low-level mesoporosity.

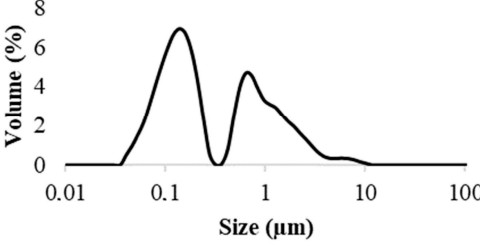

**Figure 3.** DLS for SAPO-34 sample.

The textural properties in Table 3 show that the samples prepared have a large amount of micropores with a high BET area ($S_{BET}$) and a $V_{mic}$ comparable to those in the literature

for SAPO-34. Furthermore, the low external area in the textural properties is due the nanosized crystal structure [30–32]. Figures 3 and 4 show the DLS and SEM analysis, respectively, for the SAPO-34 sample to characterize the crystal size and distribution.

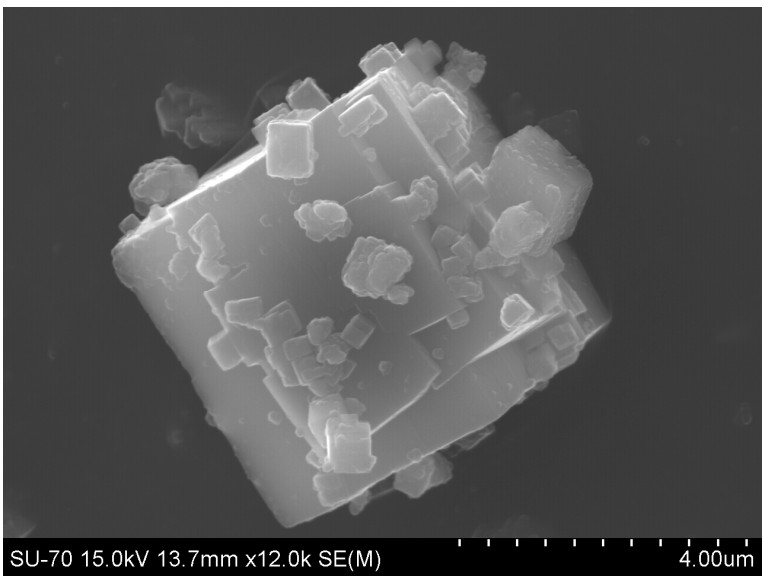

**Figure 4.** SEM for SAPO-34 sample.

The results obtained through Dynamic Light Scattering (DLS) show a range of crystal size distribution with two main sizes, related to 100 nm and 1 μm, respectively. In addition, the average particle size obtained was 200 nm. This size is suitable to be used as filler for the mixed matrix membranes without causing a high agglomerate of crystals during the polymer dissolution promoted by crystals ≤100 nm, as discussed by Wu et al. 2019 [6].

Figure 5 presents the SEM analysis for the SAPO-34 sample with an agglomerate of particles, as described by the DLS analysis, a range of particle sizes with an average of nanosized crystals can be observed by an agglomerate of cubic shaped crystals.

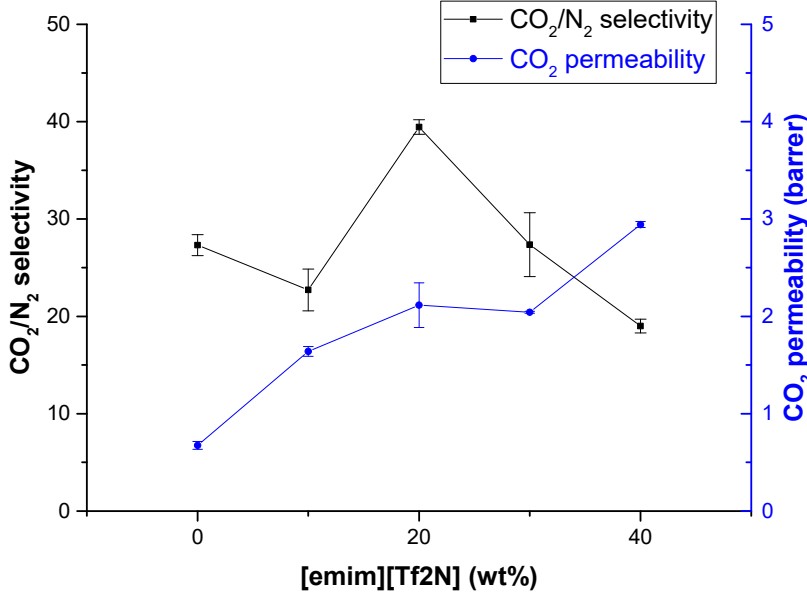

**Figure 5.** Comparison between $CO_2$ permeability, $N_2$ permeability, and $CO_2/N_2$ selectivity according to [emim][Tf$_2$N] content.

### 2.2. Gas Permeation Tests

The films were submitted to the permeation test and the permeability and selectivity obtained using Equations (1) and (2). Table 4 and Figure 5 present the average value of permeability and selectivity for each film prepared. Figure 6 exemplifies the interaction between [emim][Tf$_2$N] and zeolite surface. Figure 7 presents a comparison of the results obtained in this work with previous results in the literature organized in Table 1 for $CO_2/N_2$ separation using SAPO-34 in mixed matrix membranes plotted in a Robeson upper bound graph.

**Table 4.** Comparison in separation performance of PES/SAPO-34 and PES/SAPO-34/[emim][Tf$_2$N] membranes.

| Sample | $\alpha CO_2/N_2$ | $PCO_2$ (barrer) | $PN_2$ (barrer) |
|---|---|---|---|
| Neat PES | 21.35 ± 3.62 | 1.04 ± 0.003 | 0.05 ± 0.0090 |
| PES/SAPO-34 | 27.31 ± 1.08 | 0.67 ± 0.04 | 0.025 ± 0.0005 |
| PES/SAPO-34/[emim][Tf2N]10 | 22.71 ± 2.14 | 1.64 ± 0.05 | 0.072 ± 0.009 |
| PES/SAPO-34/[emim][Tf2N]20 | 39.44 ± 0.75 | 2.11 ± 0.23 | 0.054 ± 0.007 |
| PES/SAPO-34/[emim][Tf2N]30 | 27.36 ± 3.28 | 2.04 ± 0.01 | 0.075 ± 0.020 |
| PES/SAPO-34/[emim][Tf2N]40 | 19.00 ± 0.70 | 2.94 ± 0.03 | 0.155 ± 0.007 |

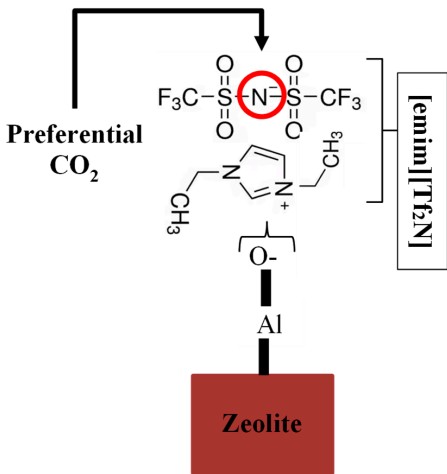

**Figure 6.** Preferential adsorption of $CO_2$.

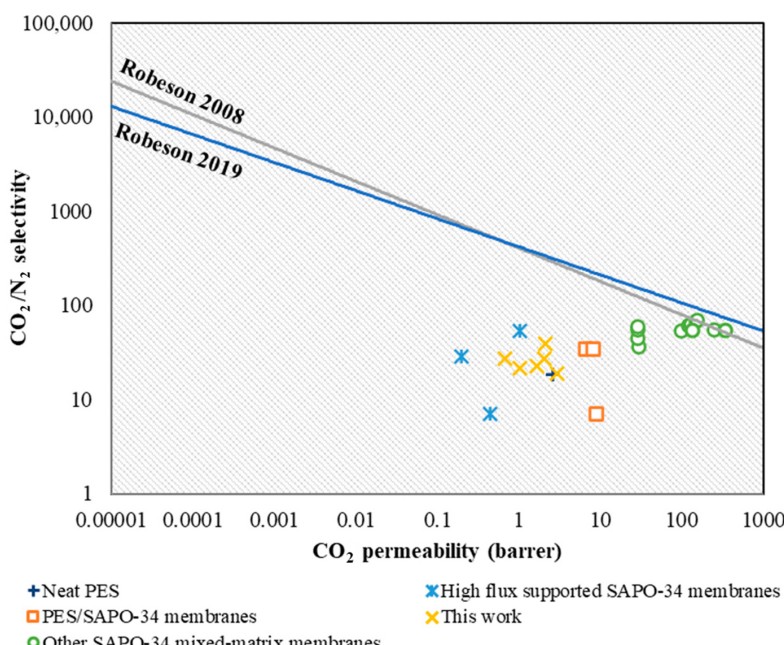

**Figure 7.** Robeson upper bound to compare different results for membranes using SAPO-34 [1,8].

In Table 4, when comparing the neat PES and PES/SAPO-34 membranes, a reduction in $CO_2$ and $N_2$ permeabilities is noted. The addition of zeolites with molecular sieving properties causes an increase in selectivity due to their intrinsic properties, while the permeabilities of gases decrease through the increase in chain rigidification, extended diffusion pathways of penetrants through the membrane or blockage of zeolite pores.

The incorporation of different amounts of [emim][Tf₂N] in the PES/SAPO-34 membrane presented an increase in $CO_2$ permeability due the preferential site for $CO_2$ adsorption when using [emim][Tf₂N], as presented in Figure 6, helping the migration of $CO_2$ throughout the membrane. The balance between polymer and plasticizer can result in a higher selectivity when compared to the PES/SAPO-34 membrane. In Table 4 and Figure 5, [emim][Tf₂N] improved the separation performance through helping to disperse the zeolite crystals more efficiently through the membrane and anchor the zeolite particles on the membrane, allowing $CO_2$ to permeate easily while still retaining $N_2$ efficiently. That behaviour can be observed in $CO_2/N_2$ selectivity for 10 wt% and 20 wt% of [emim][Tf₂N]. This behaviour is explained by Maeda et al., 1987, who evaluated the influence of additives in polymeric and mixed matrix membranes. The use of certain additives in low concentrations, of the order of 10 to 20% by weight, delays the segmental motions of the polymer accompanied by a suppression of the secondary relaxation mechanisms characteristic of polymers like polysulphone due to a loss in free volume; this phenomenon is called antiplasticization. Mahajan et al., 2002, later observed the same tendency [33,34]. This loss volume led to a higher blockage of $N_2$ due its bigger kinetic diameter when compared to $CO_2$. Moreover, the use of an additive with high $CO_2$ adsorption promoted an increase in $CO_2$ permeability, even through antiplasticization phase.

A further increase in the amount of [emim][Tf₂N], 30–40 wt%, affects the polymer chain mobility, which increases the permeability of all species, reducing the selectivity. With that, the optimal condition stablished for this membrane composition is the addition of 20 wt% of [emim][Tf₂N], which presents the best balance between an increase in the permeability of species while increasing the selectivity.

The chain rigidification can be analysed by the Glass Transition Temperature ($T_g$) presented and explained further in Section 2.3, while dealing with the characterization of the MMMs obtained. However, the addition of molecular sieving materials extends the diffusion pathway in all cases, if the molecular sieve mechanism is the limiting process, since it forces the molecules to travel a longer path than in the neat polymer membrane. This longer path through the polymer (solution–diffusion mechanism) and the material pore structure (molecular sieve mechanism) results in a longer permeation time, leading to a lower permeability, although the properties of the molecular sieve material led to an increase in selectivity by retaining larger molecules more efficiently. This affirmation can be observed when comparing the results for neat PES and PES/SAPO-34 due to the reduction in permeability for all species. Moreover, a pore blockage can be caused by the interaction between the polymer chain or additives and the particle surface reducing the accessibility of the pores, affecting the permeability of species according to the kinect diameter of the gases used [22,35].

The Robeson upper limit graph presented in Figure 7 shows that the PES/SAPO/[emim][Tf₂N]20 membrane prepared presented a higher $CO_2/N_2$ selectivity than the PES/SAPO-34 membranes in the literature due the inclusion of [emim][Tf₂N], which helped in the compatibility between zeolite and polymeric media in the optimal condition of 20 wt% of [emim][Tf₂N]. Moreover, the membranes prepared presented a lower $CO_2$ permeability than that presented in the literature. This fact derives from the reduction in interfacial voids through the use of strategies to mitigate this factor, mainly by using a drying methodology with temperature near the $T_g$ of the membranes, as explained by Mahajan, 2002 [34], and the use of [emim][Tf₂N], which reduced uncontrolled permeation across the membrane.

## 2.3. Characterization of Mixed Matrix PES/SAPO-34 Membranes

The PES/SAPO-34 membranes were prepared according to the methodology used to denote the influence of [emim][Tf₂N] on the separation performance. According to the gas permeation results obtained, the addition of 20 wt% of [emim][Tf₂N] in membrane

fabrication promotes the best ideal separation performance in this composition. DMTA and TGA analyses for PES, PES/SAPO-34 membrane, and PES/SAPO-34/[emim][Tf$_2$N]20 are depicted in Figures 8 and 9, respectively.

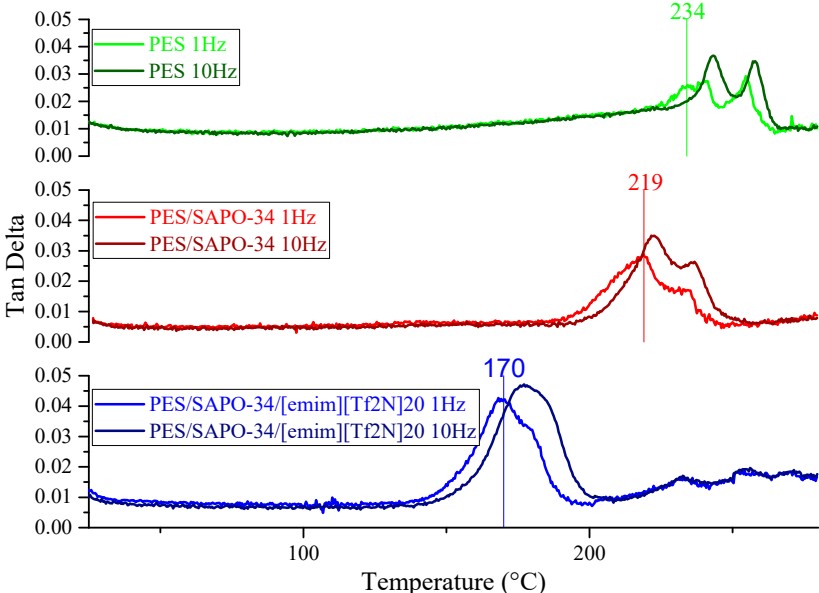

**Figure 8.** DMTA analysis for PES, PES/SAPO-34, and PES/SAPO-34/[emim][Tf$_2$N]20 at 1 Hz and 10 Hz.

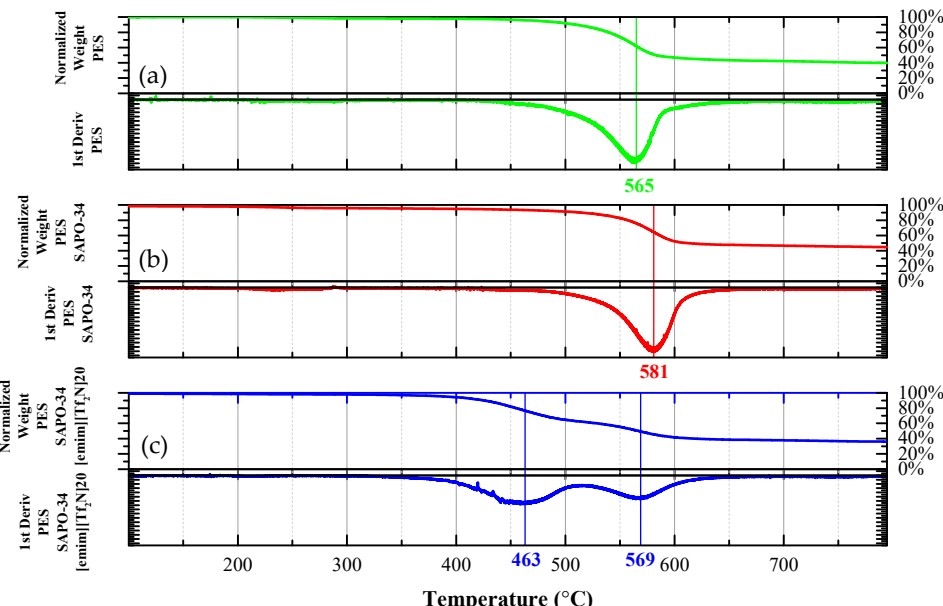

**Figure 9.** TGA analysis for (**a**) PES, (**b**) PES/SAPO-34, and (**c**) PES/SAPO-34/[emim][Tf$_2$N]20.

The DMTA analysis, in Figure 8, presents the Glass Transition Temperature (T$_g$) at a 1 Hz frequency of 234 °C, 219 °C, and 170 °C for PES, PES/SAPO-34, and PES/SAPO-34/[emim][Tf$_2$N]20, respectively. Those temperature express the flexibility degree of the membranes, showing that the inclusion of SAPO-34 does not lead to a rigidified polymer/zeolite interface, since those rigidified interfaces would lead to a higher Tg than neat PES. Moreover, the inclusion of [emim][Tf$_2$N] reduces chain rigidity and improves flexibility, lowering the membrane T$_g$ to better accommodate with the solvent evaporation. A lower T$_g$ expresses a more flexible membrane, which can better accommodate rigid particles when compared to a more rigid membrane [34,36].

Furthermore, the TGA analysis in Figure 9 shows that the degradation temperature for PES is 565 °C, while the one for the PES/SAPO-34 is 581 °C. This is due to the addition of the zeolite particle improving the thermal stability of the membrane. Moreover, the degradation temperature for the PES/SAPO-34/[emim][Tf$_2$N]20 membrane presents the first degradation temperature in 463 °C for [emim][Tf$_2$N] [37–40] with a subsequent degradation of PES at 569 °C, which is comparable to neat PES. [emim][Tf$_2$N] functions as a linking agent to reduce the stress from the interaction between the SAPO-34 crystals and bulk PES as explained by Mahajan et al., 2002 [34]. Figure 10 presents the FTIR analysis for neat PES, PES/SAPO-34, PES/SAPO-34/[emim][Tf$_2$N]20, and [emim][Tf$_2$N].

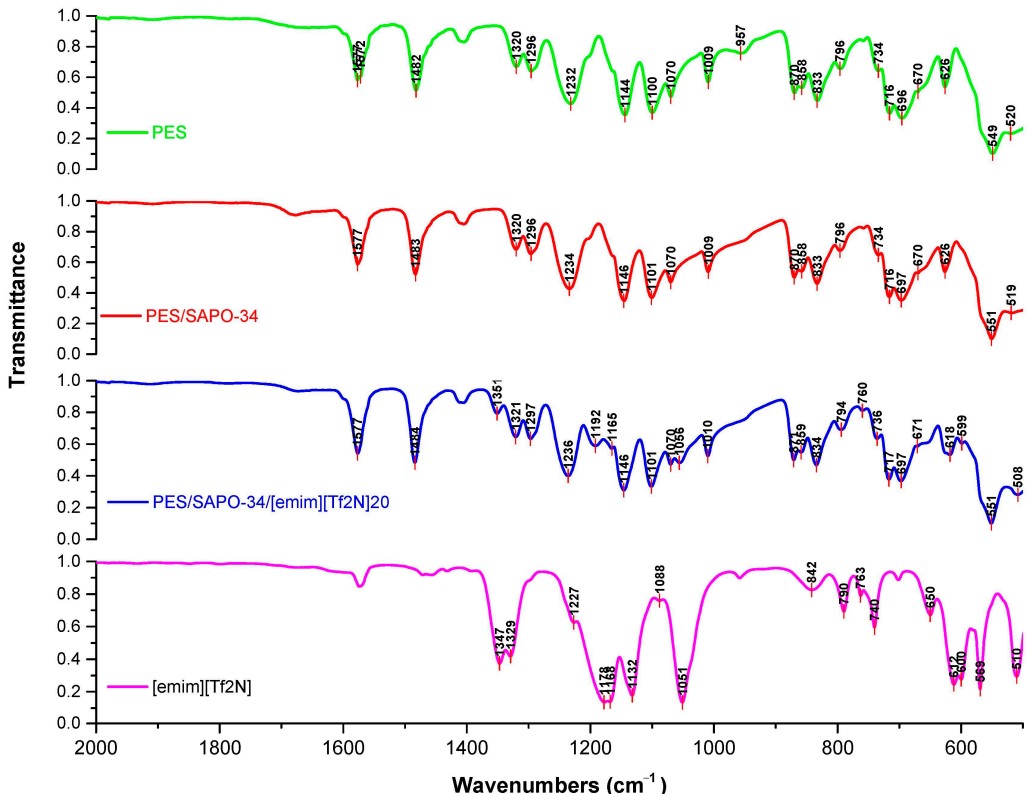

**Figure 10.** FTIR for neat PES, PES/SAPO-34, PES/SAPO-34/[emim][Tf$_2$N], and [emim][Tf$_2$N].

The three peaks observed at 1576–1577 cm$^{-1}$, 1482–1484 cm$^{-1}$, and 1405–1406 cm$^{-1}$ indicate the presence of benzene rings. The two peaks at 1320–1321 cm$^{-1}$ and 1296–1297 cm$^{-1}$ indicate the presence of ether function. The two peaks at 1144–1146 cm$^{-1}$ and 1100–1101 cm$^{-1}$ indicate the presence of a sulfone group. All spectra are in accordance with the FTIR spectra for PES in the literature [41,42] and implies that the addition of SAPO-34 or the use of ILs does not change the membrane matrix.

Moreover, the appearance of peaks in 1351 cm$^{-1}$, 1192 cm$^{-1}$, 1055 cm$^{-1}$, and others dislocated peaks like 618.3 cm$^{-1}$ and 508.4 cm$^{-1}$ in the PES/SAPO-34/[emim][Tf$_2$N]20 membrane can be related to the inclusion of [emim][Tf$_2$N] in the membrane, since those peaks also appear in the FTIR for [emim][Tf$_2$N] substance. However, it is notorious in Figure 10 that the inclusion of [emim][Tf$_2$N] presents no influence in the overall polymer chain structure.

A contact angle analysis was conducted in ambient conditions and is presented in Table 5 with the average values for neat PES, PES/SAPO-34, and PES/SAPO-34/[emim][Tf$_2$N]20, respectively.

**Table 5.** Average contact angle.

| Sample | Average Contact Angle (°) |
|---|---|
| PES | 70 ± 2 |
| PES/SAPO-34 | 71 ± 2 |
| PES/SAPO-34/[emim][Tf2N]20 | 61 ± 2 |

All membranes show a slightly hydrophilic characteristics, which is in accordance with the literature [43]. This is more pronounced with the addition of the [emim][Tf$_2$N] due to its hydrophilicity. It is a drawback in gas separation due to the adsorption of water that can decrease the membrane separation efficiency, degrade the zeolite, and reduce the operation time of the membrane.

An SEM analysis for neat PES, PES/SAPO-34, and PES/SAPO-34/[emim][Tf$_2$N]20 membranes is presented in Figures 11–13, respectively, and a zoom in on the particles in Figure 14 for a better understanding of the particle dispersion.

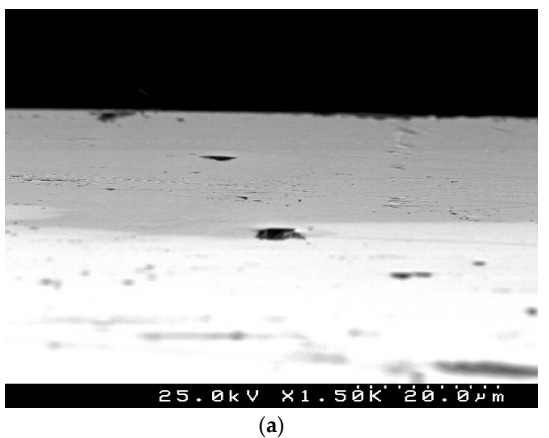
(**a**)

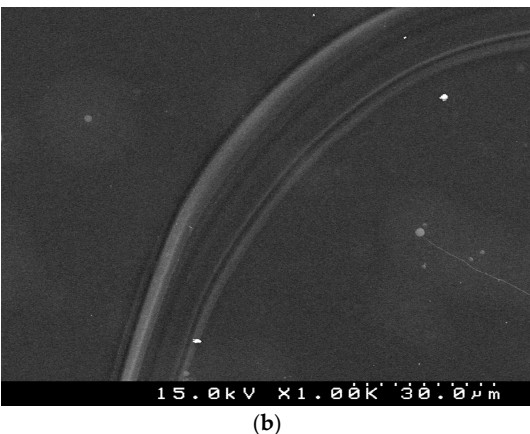
(**b**)

**Figure 11.** (**a**) Cross-section and (**b**) surface of neat PES membrane.

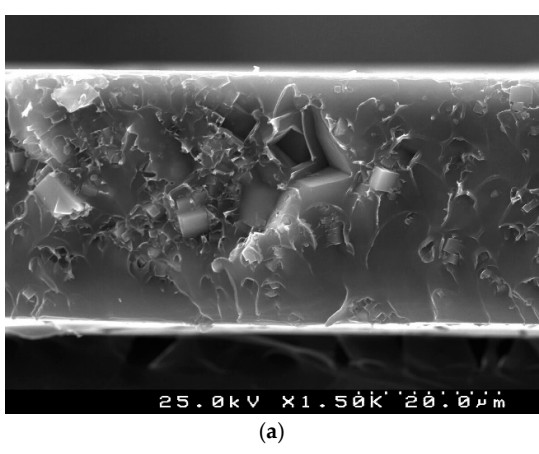
(**a**)

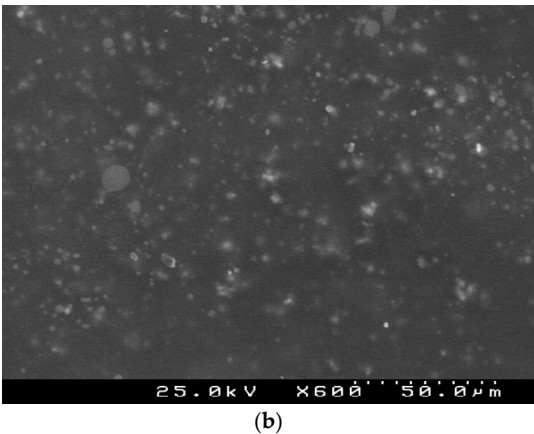
(**b**)

**Figure 12.** (**a**) Cross-section and (**b**) surface of PES/SAPO-34 membrane.

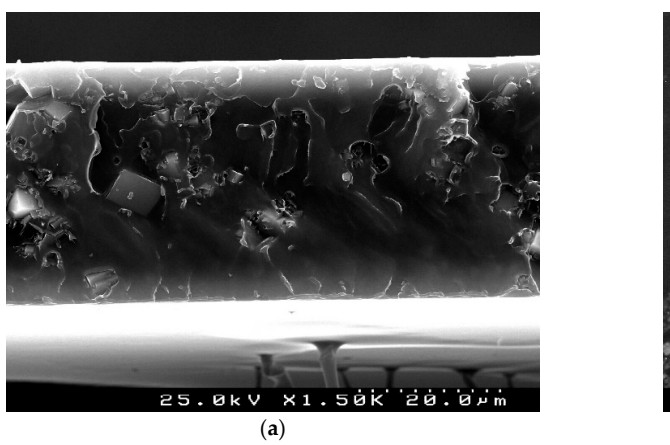
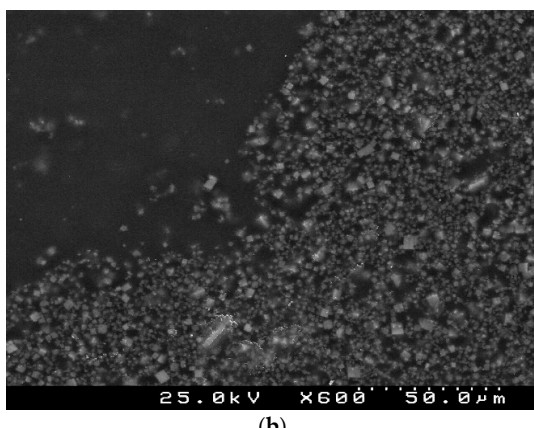

(**a**)          (**b**)

**Figure 13.** (**a**) Cross-section and (**b**) surface of PES/SAPO-34/[emim][Tf$_2$N]20 membrane.

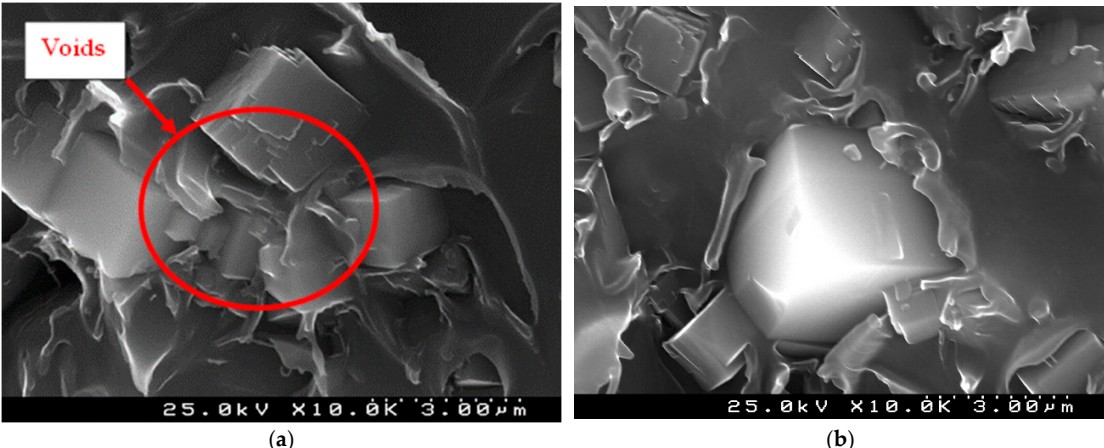

(**a**)          (**b**)

**Figure 14.** Zoom in particle distribution for (**a**) PES/SAPO-34 and (**b**) PES/SAPO-34/[emim][Tf$_2$N]20 membranes.

Figure 11a present the SEM analysis of the neat PES membrane cross-section and surface. The dense PES membrane presents no imperfections and a similar homogeneity through the membrane. Figure 12a presents the SEM analysis of the PES/SAPO-34 membrane cross-section, and the agglomeration of crystals along the polymeric matrix could allow gases to permeate through the interface of the crystals. Figure 13a presents the SEM analysis of the PES/SAPO-34/[emim][Tf$_2$N]20 membrane cross-section with more disperse crystals in the polymeric matrix. Figures 12b and 13b present a similar surface for both membranes. Figure 14a presents a zoom in of the particle agglomeration, which could cause a reduction in selectivity due the presence of voids between particles. The use of [emim][Tf$_2$N] helped to reduce crystal agglomeration, as presented in Figure 14b. Higher dispersion diminishes imperfections and increases the separation performance of the membrane.

Table 4 and Figure 5 combined with the SEM analysis in Figure 14 show how a well-dispersed filler can affect the membrane separation characteristics and how the use of ionic liquids can improve the separation performance due to an increase in the interaction between the zeolite and polymer, the reduced formation of aggregates, increased dispersion of the zeolite in the polymeric matrix, increased membrane flexibility to better accommodate particles, and preferential interaction with $CO_2$ to increase permeability.

## 3. Materials and Methods

### 3.1. Materials

The polymer used as the media for the membrane in this study was polyethersulfone (PES) from Goodfellow, transparent, with a 3 mm granular size and molecular weight of

58,000 g/mol. The inorganic filler SAPO-34 was prepared using aluminium isopropoxide ($\geq$99% purity), phosphoric acid (85%), colloidal silica HS-40, morpholine ($\geq$99% purity), and tetraethylammonium hydroxide (TEAOH) solution (35 wt% in water) purchased from Sigma-Aldrich (St. Louis, MO, USA). PES and SAPO-34 were dried overnight at 105 °C and 200 °C, respectively, prior to use. The solvent used for membrane preparation and casting was N-methyl-pyrroline (NMP) ($\geq$99.8% purity) from Roth. The ionic liquid used was [emim][Tf$_2$N] ($\geq$98%) from Sigma-Aldrich was selected for its high thermal stability and CO$_2$ adsorption characteristics. It presents a decomposition temperature of 360 °C and was used without further purification.

### 3.2. Synthesis and Characterization of SAPO-34

The SAPO-34 sample was synthesized using a gel with the molar composition 1.0 Al$_2$O$_3$: 1.0 P$_2$O$_5$: 0.6 SiO$_2$: 1.5 Morpholine: 0.5 Tetraethylammonium hydroxide (TEAOH): 70 H$_2$O. The zeolite was prepared using a dry-gel methodology with 1:1 mass ratio of dry mass/water, heated at 200 °C for 24 h; this condition was determined by a previous study [44]. The produced mixture was then centrifuged, washed, dried at 105 °C in an oven, and calcined at 560 °C for 8 h. The samples were characterized via Powder X-Ray Diffraction (PXRD) in a PANalytical Empyrean diffractometer (PANalytical, Malvern, UK) using a Bragg–Brentano geometry and CuK$\alpha$ X-radiation, an Energy Dispersive X-ray spectrometry (EDS) on a Hitachi SU-70 microscope (Hitachi, Tokyo, Japan) equipped with EDS system, laser diffraction spectroscopy (LDS) in a Malvern Masterziser 2000 from Malvern Instruments (Malvern, UK) and N$_2$ adsorption and desorption in a NOVAtouch LX4 from Quantachrome (Boynton Beach, FL, USA).

### 3.3. Synthesis of Mixed Matrix PES/SAPO-34 Membranes

PES/SAPO-34 membranes were prepared according to the mass composition of 20 wt% of PES and 20 wt% of SAPO-34, based on polymer mass, in NMP. The SAPO-34 zeolite was added and stirred with N-methyl-2-pyrrolidone (NMP) for 1 h and in an ultrasonic bath for 4 h. Then, a small quantity of PES was added to the NMP/SAPO-34 solution and mixed for 1 h at 80 °C. The remaining PES was added and mixed for 3 h at 80 °C, and the solution was stirred overnight at room temperature. The solutions were degassed for 4 h in an ultrasonic bath and casted with a knife of 300 μm of initial thickness in a glass plate, dried at 80 °C for 2 h under vacuum and at 200 °C for 20 h under vacuum. The PES/SAPO-34/[emim][Tf$_2$N] membrane synthesis followed the same procedure using 10, 20, 30, and 40%w/w, respectively, of IL added in the NMP-SAPO-34 solution step.

### 3.4. Gas Permeations Tests

Gas permeation experiments were conducted using pure gases to determine their ideal permeability and diffusivity coefficients and the results were compared to each other in a permeability unit called Barrer ($10^{-10}$ cm$^3_{STP}$ × cm/cm$^2$ × s × cmHg). A schematic representation of the permeation system is shown in Figure 15 The membranes were cut into 47 mm disks and mounted on a steel permeation cell and evaluated at feed pressure of 10 bar at room temperature. The ideal permeabilities were obtained according to Equation (1).

$$P_i = \frac{V_s l}{T_{AMB} \Delta p} \frac{T_{STP}}{p_{STP} A} \frac{dp}{dt} \tag{1}$$

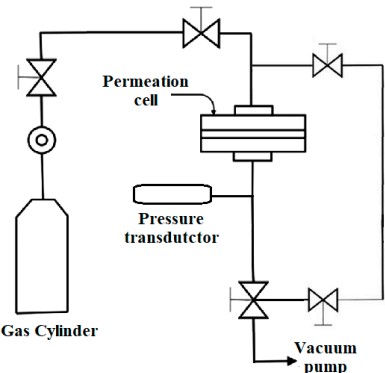

**Figure 15.** Schematics of the permeation system.

The permeability of a certain gas ($P_i$) is related to the thickness of the film ($l$), the fixed volume of the permeate ($Vs$), which is measured as standard. The ambient temperature ($T_{AMB}$), the difference of pressure between permeate and retentate ($\Delta p$), the permeation area ($A$), and temperature and pressure in normal conditions ($T_{STP}$ and $P_{STP}$). The ideal selectivity results ($\alpha_{ab}$) are represented by the ratio between the permeability ($P$) of two pure gases, *a* and *b*, respectively, measured separately under the same conditions, in the same membrane to evaluate the separation performance, as presented in Equation (2).

$$\alpha_{ab} = \frac{P_a}{P_b} \tag{2}$$

*3.5. Characterization of Mixed Matrix PES/SAPO-34 Membranes*

The samples were characterized by a Dynamic Mechanical and Thermal Analysis (DMTA) in Triton Technology equipment, Leicestershire, UK, model Tritec 2000, with heating rate of 2 °C/min in 1 and 10 Hz frequencies for all samples to verify the glass transition temperature, a Thermogravimetric Analysis (TGA) in a TA Instruments TGA Q500 with an $N_2$ flow of 10 mL/min at 10 °C/min to obtain the degradation temperatures, and Fourier-transform infrared spectroscopy (FTIR) was used to compare the presence or absence of chemical components and alterations. A total attenuated reflectance non-destructive technique (ATR) was used in the range of 4000–550 cm$^{-1}$ in a Perkin Elmer spectrometer model Spectrum Two FT-IR with detector DTGS and a KBr beam-splitter, resolution of 4.0 cm$^{-1}$, 64 accumulations, and 50 N of constant applied force, Scanning Electron Microscopy (SEM) on a Hitachi S-4100 microscope to compare the membrane structure, morphology, and zeolite dispersibility, and a Contact Angle analysis using a Dataphysics equipment, model Contact Angle System OCA, from Filderstadt, Germany.

**4. Conclusions**

To study the influence of ionic liquids as additives into mixed matrix membranes, [emim][Tf$_2$N] was added in situ to prepare similar mixed matrix membranes. Concentrations varying from 10 to 40 wt%, according to polymer mass, were used in the membrane preparation to compare the influence of [emim][Tf$_2$N] concentrations in MMM and obtain an optimal condition.

The results showed that PES/SAPO-34 membrane containing 20 wt% of [emim][Tf$_2$N] presented the highest separation performance among the tested membranes due to an increase in the dispersion of the crystals in the polymeric matrix, helped by a better interaction of zeolite–polymer, as well as keeping the interaction during the solvent removal, while promoting an increase in $CO_2$ permeation and the selectivity of the PES/SAPO-34/[emim][Tf$_2$N]20 membrane. This increase in separation performance is due the reduction in $N_2$ permeability by the antiplasticization state, which reduces the polymer chain mobility and free volume. Since this antiplasticization phase occurs at low concentrations of additives (10–20 wt%), the permeation of species in the membrane is even more regu-

lated by kinetic diameter, and the use of a high $CO_2$ solubility additive explains how $CO_2$ permeation was not affected. However, a further increase in [emim][Tf$_2$N] in PES/SAPO-34 mixed matrix membrane, 30 wt% or higher, resulted in the higher permeation of all species due to the high influence in the polymer chain mobility and, consequently, a reduction in $CO_2$/$N_2$ selectivity.

Moreover, the use of ionic liquid does not affect the general structure of the polymer, as presented by tge FTIR and contact angle analysis, only delaying the segmental motion of the polymer and the suppression of the secondary relaxation mechanism and free volume with low concentrations of additive. Furthermore, it keeps the thermal resistance, as observed with TGA characterization. However, [emim][Tf$_2$N] promoted more stable interaction and increased dispersion of the zeolite in the polymer, as can be observed in the SEM analysis, while it improved the membrane flexibility related to a reduction in the membrane $T_g$ when compared to the neat PES and PES/SAPO-34 membranes.

In addition, other ionic liquids and deep eutectic solvents are also planned to be evaluated and compared as more eco-friendly and less expensive alternatives, to understand the effects of those in the separation performance of PES/SAPO-34.

**Author Contributions:** Conceptualization, J.S.C.; Z.L.; P.B. and L.M.G.-F.; investigation, J.S.C.; resources, Z.L.; P.B. and L.M.G.-F.; writing—original draft preparation, J.S.C.; writing—review and editing, J.S.C.; Z.L.; P.B. and L.M.G.-F.; visualization, J.S.C.; supervision, Z.L.; P.B. and L.M.G.-F.; project administration, Z.L.; P.B. and L.M.G.-F.; funding acquisition, J.S.C.; Z.L.; P.B. and L.M.G.-F. All authors have read and agreed to the published version of the manuscript.

**Funding:** The authors gratefully acknowledge the fundings from the Strategic Project of CIEPQPF (UIDB/00102/2020), CICECO-Aveiro Institute of Materials (UIDB/50 011/2020, UIDP/50 011/2020 & LA/P/0 0 06/2020), CIMO (UIDB/00690/2020 and UIDP/00690/2020) and SusTEC (LA/P/0007/2021), financed by Fundação para a Ciência e Tecnologia (FCT) through national funds. J. S. Cardoso is also grateful for the financial support of the FCT through the PhD grant (SFRH/BD/148170/2019).

**Data Availability Statement:** All data underlying the results are available as part of the article and no additional source data are required.

**Acknowledgments:** The authors gratefully acknowledge the support from CIEPQPF, University of Coimbra, CICECO—Aveiro Institute of Materials, CIMO—Montain Research Centre, SusTEC—Associate Laboratory for Sustainability and Technology in Mountains Regions and FCT—Fundação para a Ciência e Tecnologia.

**Conflicts of Interest:** The authors declare no conflict of interest.

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
