# Peer review of "The Functionalization of PES/SAPO-34 Mixed Matrix Membrane with [emim][Tf2N] Ionic Liquid to Improve CO2/N2 Separation Properties"

_inorganics, doi:10.3390/inorganics11110447_

Round 1
Reviewer 1 Report
Comments and Suggestions for Authors
The effect of ionic liquid [emim][Tf2N] as an additive on the separation of CO2 and N2 over PES/SAPO-34 mixed matrix membrane was studied. Enhanced selectivity as well as CO2 permeability was obtained by proper amount of [emim][Tf2N]. However, the role of [emim][Tf2N] is still not clear in the text. Especially, why the CO2 permeability was increased while N2 permeability decreased from PES/SAPO-34/[emim][Tf2N]10 to PES/SAPO-34/[emim][Tf2N]20. Thus, major revisions are needed before it can be published. The followings are some other suggestions:
1) Fig.7 and Fig.8 should be interchange since Fig.8 is discussed ahead of Fig.7.
2) Why reference 20 is cited in Fig.7? Preferential adsorption of CO2 is not discussed in that reference.
3) Si/Al ratio of obtained SAPO-34 should be given since it may affect CO2/N2 selectivity.
Author Response
Reviewer 1
The effect of ionic liquid [emim][Tf2N] as an additive on the separation of CO2 and N2 over PES/SAPO-34 mixed matrix membrane was studied. Enhanced selectivity as well as CO2 permeability was obtained by proper amount of [emim][Tf2N]. However, the role of [emim][Tf2N] is still not clear in the text. Especially, why the CO2 permeability was increased while N2 permeability decreased from PES/SAPO-34/[emim][Tf2N]10 to PES/SAPO-34/[emim][Tf2N]20. Thus, major revisions are needed before it can be published. The followings are some other suggestions:
A: We thank the reviewer for the corrections and proceeded to change it accordingly. We proceeded to add a more insightful explanation about the increase in CO2 permeability with reduction in N2 permeability in concentrations between 10-20wt% in Section 3.2 based on the literature.
1) Fig.7 and Fig.8 should be interchange since Fig.8 is discussed ahead of Fig.7.
A: We thank the reviewer for the corrections and proceeded to change it accordingly. We changed this paragraph position and reorganized this section, since we think it is more efficient to present and explain DMTA and TGA analysis together.
2) Why reference 20 is cited in Fig.7? Preferential adsorption of CO2 is not discussed in that reference.
A: We thank the reviewer for the corrections and proceeded to change it accordingly.
3) Si/Al ratio of obtained SAPO-34 should be given since it may affect CO2/N2 selectivity.
A: We thank the reviewer for the corrections and proceeded to change it accordingly. We added in Table 2 the XRF result and a discussion about the Si/Al ratio and its importance for CO2 adsorption.

Reviewer 2 Report
Comments and Suggestions for Authors
This manuscript “Functionalization of PES/SAPO-34 mixed matrix membrane with [emim][Tf2N] ionic liquid to improve CO2/N2 separation properties” deals with the modification of PES membrane with nano-sized silico-aluminophos-17 phate-34 (SAPO-34) and ionic 16 liquid [emim][Tf2N] for the gas separation. Although extensive work has been performed, several points must be improved before acceptance of this manuscript.
1) The abstract should be rewritten. The author should tell the simple results and meanings.
2) All abbreviations must be spelled out the first time they are mentioned (for example, PSA, TSA).
3) The introduction should be rewritten. A comparison with other works is needed. The description of Table 1 in the text is needed.
4) The novelty of this research is needed in the last paragraph of the introduction.
5) What the support was used to prepare polymer film? Glass support or other?
6) How do the authors compare SAPO-34 sample with CHA standard in Figure 2? For which compound is this Chabazite (CHA) standard?
7) A greater description of the results obtained in the text is needed, and not just Figures and Tables. This applies to the entire section “3.1. SAPO-34 synthesis and characterization”.
8) The text does not contain a detailed description of Figure 5. Compare the results obtained in the study with the literature data in the text (State-of-art). Why are your membranes better than those presented in Table 1?
9) Why does the selectivity of separation decrease after a 20% addition of modifier? Description required in the text of the manuscript with the literature confirmation.
10) The peaks related to the MMMs need to be referenced in Figure 10.
11) Figures 11-13 contains SEM micrographs for surface with different magnifications that is not possible to compare (for neat PES membrane and modified membranes). The authors must use the same magnification micrographs and combined in one Figure.
12) Additional SEM micrographs of particles outside the polymer film are required (Figure 14).
13) The author should tell the simple results in Conclusion.
Comments on the Quality of English LanguageMinor editing of English language required
Author Response
Reviewer 2
This manuscript “Functionalization of PES/SAPO-34 mixed matrix membrane with [emim][Tf2N] ionic liquid to improve CO2/N2 separation properties” deals with the modification of PES membrane with nano-sized silico-aluminophos-17 phate-34 (SAPO-34) and ionic 16 liquid [emim][Tf2N] for the gas separation. Although extensive work has been performed, several points must be improved before acceptance of this manuscript.
1) The abstract should be rewritten. The author should tell the simple results and meanings.
A: We thank the reviewer for the corrections and proceeded to change it accordingly. We reformulated part of the Abstract in accordance with the Conclusion.
2) All abbreviations must be spelled out the first time they are mentioned (for example, PSA, TSA).
A: We thank the reviewer for the corrections and proceeded to change it accordingly.
3) The introduction should be rewritten. A comparison with other works is needed. The description of Table 1 in the text is needed.
A: We thank the reviewer for the corrections and proceeded to change it accordingly. We reworked the Introduction to lead for a deeper and insightful explanation about the use of [emim][Tf2N] and its role in mixed matrix membranes. Moreover, Table 1 is used as example of surface modifiers in PES/SAPO-34 membranes in literature but it is also used as comparison criteria for the membranes prepared in this work, which those literature results are presented in the Robeson upper limit in Figure 6 as PES/SAPO-34 membranes.
4) The novelty of this research is needed in the last paragraph of the introduction.
A: We thank the reviewer for the corrections and proceeded to change it accordingly. We added a final paragraph in Introduction related to the novelty of this work, an explanation on why we obtained this optimal point and how the use of low amounts of additive can lead to an antiplasticization effect that can be benefic to increase separation performance.
5) What the support was used to prepare polymer film? Glass support or other?
A: We thank the reviewer for the corrections and proceeded to change it accordingly. We added in Section 2.3 the use of a glass plate to prepare the membranes.
6) How do the authors compare SAPO-34 sample with CHA standard in Figure 2? For which compound is this Chabazite (CHA) standard?
A: We thank the reviewer for the corrections and proceeded to change it accordingly. We compared the XRD patterns of CHA and SAPO-34 sample since both presents the same structure with same planes of dispersion, this methodology is the standard by the International Zeolite Association (IZA).
7) A greater description of the results obtained in the text is needed, and not just Figures and Tables. This applies to the entire section “3.1. SAPO-34 synthesis and characterization”.
A: We thank the reviewer for the corrections and proceeded to change it accordingly. We reworked the Section 3.1 to promote a further explanation in the SAPO-34 synthesis and characterization topic.
8) The text does not contain a detailed description of Figure 5. Compare the results obtained in the study with the literature data in the text (State-of-art). Why are your membranes better than those presented in Table 1?
A: We thank the reviewer for the corrections and proceeded to change it accordingly. We added a description of Figure 5 (now Figure 6) and a comparison paragraph between the membranes fabricated in this work and the ones presented in literature.
9) Why does the selectivity of separation decrease after a 20% addition of modifier? Description required in the text of the manuscript with the literature confirmation.
A: We thank the reviewer for the corrections and proceeded to change it accordingly. We included an explanation about the behavior observed in concentrations between 10-20% with some literature support. We also reorganized Section 3.2 to a more fluid explanation of the results obtained.
10) The peaks related to the MMMs need to be referenced in Figure 10.
A: We thank the reviewer for the corrections and proceeded to change it accordingly. We added the label for peaks to illustrate what is explained in the following paragraph. The peaks for all membranes are presented in the FTIR image (Figure 10) showing that all membranes presented the same spectrum and the inclusion of [emim][Tf2N] does not affected the membrane general structure.
11) Figures 11-13 contains SEM micrographs for surface with different magnifications that is not possible to compare (for neat PES membrane and modified membranes). The authors must use the same magnification micrographs and combined in one Figure.
A: We thank the reviewer for the corrections and proceeded to change it accordingly. We changed Figure 11b for another with higher magnification, however we only have a 15kV, x1.00K, 30.0 µm magnification for neat PES membrane, we hope this is enough to compare this image with the others provided for PES/SAPO-34 and PES/SAPO-34/[emim][Tf2N]20 surface images. For cross-section SEM images, all of those were obtained in magnification of 25kV, x1.50K, 20.0 µm since this magnification is still in range with the surface SEM images but providing enough details for the particles.
12) Additional SEM micrographs of particles outside the polymer film are required (Figure 14).
A: We thank the reviewer for the corrections and proceeded to change it accordingly. We provided a SEM micrograph of the SAPO-34 particles outside the polymer film as presented in Figure 5. We decided to place it with the DLS analysis which relate to the particle size analysis.
13) The author should tell the simple results in Conclusion.
A: We thank the reviewer for the corrections and proceeded to change it accordingly. We reformulated part of the Conclusion to a better explanation on the antiplasticization effect and the influence of [emim][Tf2N] in the membrane structure, which we hope lead to an improved understanding of the results obtained in this work.

Round 2
Reviewer 1 Report
Comments and Suggestions for Authors
The paper can be published after revision.
Reviewer 2 Report
Comments and Suggestions for Authors
The revised paper can be accepted in present form.